# Gas–Liquid Interaction Characteristics in a Multiphase Pump under Different Working Conditions

**Yuxuan Deng [1,*], Xiaodong Wang [2], Jing Xu [1], Yanna Li [1], Yanli Zhang [1] and Chunyan Kuang [1]**

[1]   BaiLie School of Petroleum Engineering, Lanzhou City University, Lanzhou 730071, China
[2]   Pipechina West Pipeline Company Lanzhou Gas Transmission Branch Company, Lanzhou 730070, China
*   Correspondence: dengyuxuan@lzcu.edu.cn; Tel.: +86-159-0815-5884

**Abstract:** In this study, we analyze gas–liquid interaction characteristics using a heterogeneous two-fluid model to investigate the influence of interphase force on multiphase pump performance. Two-phase transport platforms are used in oil and gas development to eliminate the need for separation equipment and reduce costs. Full-channel numerical calculations were conducted for an axial-flow multiphase pump based on different inlet gas void fractions (IGVFs) and flow rates. The results indicate that the interaction force of each phase is relatively large in the rotor–stator interference region, and the drag, lift, virtual mass, and turbulent dispersion forces increase with an increase in IGVF or when deviating from the design condition ($Q$ = 50 m$^3$/h). The interphase forces (resistance, lift, virtual mass force, and turbulent dispersion) increase considerably in the impeller passage and minimally in the guide blade passage. Under the conditions of small and high flows, the force of each phase changes considerably in the impeller and diffuser passages, respectively. Furthermore, the turbulent kinetic energy in the flow passage corresponds to the change trend of the interphase force, indicating that the interphase force causes energy loss inside a multiphase pump. These results provide essential information for the optimization of the hydraulic design of multiphase pumps.

**Keywords:** multiphase pump; two-phase flow; inlet gas void fraction; interphase force; numerical calculation

## 1. Introduction

Deep-sea oil and gas resources usually comprise a mixture of oil, gas, and water. Utilizing separation equipment is the traditional approach to separating oil and gas, which are then transported separately using a pump and compressor. As such, this process employs with complex piping systems and separation equipment, resulting in high investment costs [1,2]. In contrast, a two-phase transport platform eliminates part of the pipeline and separation equipment, saving 40% of infrastructure investment [3]. This approach has been widely used in oil and gas development [4,5]. As the core equipment of a two-phase transport platform, the axial-flow multiphase pump ensures smooth and efficient operation and plays a key role in the development and utilization of oil and gas resources.

Many scholars have conducted research on optimization design and internal flow field analysis to improve the performance of multiphase pumps. Cao et al. proposed a design method to determine blade placement angle based on a given velocity moment [6]. Their experimental results showed that an impeller designed using this method had a wide working range. Zhang et al. proposed a hydraulic design method for impeller blades of multiphase pumps under large design flows based on a meridional surface network [7]. Kim et al. conducted variable sensitivity analysis, whereby geometric parameters were selected as optimization variables, such as the inlet width, outlet width, hub ratio, and placement angle [8]. Moreover, the impeller and diffuser were simultaneously optimized by constructing an approximate function relation between the geometric parameters, pressure rise, and efficiency under pure water conditions. The results showed that the flow

separation in the optimized pump was suppressed, and the pump efficiency increased by 3.5%. Liu et al. adopted a polynomial function to directly control the blade placement angle of a mixed-transport pump [9]. They used the blade placement angle as an optimization variable and conducted orthogonal optimization with five factors and four levels. The results showed that the pressure of the optimized pump increased by 6.69%.

Zhang et al. used numerical simulation results as samples to study the influence of inlet gas void fractions (IGVFs) on multiphase pump performance [10]. They used a backpropagation neural network to establish the response function relationship between optimization variables and optimization objectives. Finally, the optimization results of comprehensive performance under 0%, 10%, 30%, and 40% gas content conditions were obtained through multiobjective genetic algorithm optimization in variable space. Xu et al. conducted an experimental study on a spiral axial-flow gas–liquid multiphase pump [11]. They concluded that the pump could operate stably at 0–50% gas content. The head and efficiency gradually decreased with an increase in inlet gas content. Zhang et al. divided various flow forms into a bubble balloon and stratified flows according to the diameter and number of bubbles in the pump [12,13]. Saadawi et al. debugged a multiphase pump [14]. They found that increasing the speed extended the pump's operating range at 50% inlet gas. Alberto et al. observed, through visualization experiments, that increasing impeller speed resulted in more uniform gas–liquid two-phase mixing [15,16].

Some parameters cannot be obtained using the current test techniques. With the rapid development of technology in recent years, many scholars now use computational fluid dynamics software to conduct numerical research on multiphase pumps. Zhang et al. conducted a numerical analysis on an axial-flow multiphase pump under different inlet gas holdup conditions [17]. They found higher gas holdup in the area near the outlet hub of the impeller. The higher the gas content of the inlet, the higher the degree of its aggregation. Liu et al. carried out a numerical simulation on four groups of mixed over-flow media, obtaining various viscosity by mixing glycerol with different proportions of water [18]. Their analysis showed that the turbulent kinetic energy in the flow channel increased with an increase in viscosity. Zhang et al. conducted unsteady calculations of an axial-flow multiphase pump under different inlet gas holdup and flow conditions [19,20]. Their analysis showed that the main cause of pressure pulsation was rotor–stator interference and the maximum value appeared near the rotor–stator interference area. Zhang et al. also reached a similar conclusion [21]. Yu et al. preliminarily calculated and analyzed the magnitude and variation of gas–liquid interforce in a multiphase pump [22].

In summary, most of the studies on multiphase pumps have focused on performance optimization and internal flow field analysis, and few studies have been conducted on gas–liquid interaction characteristics. In particular, the mechanism of gas–liquid interaction is not well understood. In this study, we numerically calculated the internal flow field of the axial flow in a multiphase pump using a heterogeneous two-fluid model to investigate the influence of interphase force on the performance of the multiphase pump. The change rule of interphase force in a multiphase pump was studied under different IGVF (5%, 10%, 15%, 20%, and 25%) and flow rate ($Q$ = 30, 40, 50, 60, and 70 m$^3$/h) conditions to illustrate their effects on interphase force.

## 2. Numerical Model

### 2.1. Computational Method

A heterogeneous two-fluid model was used in this study to calculate the two-phase flow in a multiphase pump. Reynolds time and homogeneous Navier–Stokes equations were used to solve the calculations. The governing equations for a steady incompressible fluid are as follows:

The continuity equation:

$$\nabla \cdot (\alpha_k \rho_k V_k) = 0 \tag{1}$$

The momentum equation:

$$\nabla \cdot (\alpha_k \rho_k V_k V_k) = -\alpha_k \nabla p + \nabla \cdot (\alpha_k \tau_{ij}) + M_k + f_k \tag{2}$$

where the subscript $k$ represents any phase ($l$ is liquid, and $g$ is gas); $\alpha_k$ is the $k$-phase volume fraction, satisfying $\alpha_g + \alpha_l = 1$; $\rho_k$ is the $k$-phase density; $V_k$ is the $k$-phase absolute velocity; $p$ is the pressure; $\tau$ is the viscous stress tensor; $M_k$ is the interphase force on phase K; and $f_k$ is the mass force.

ANSYS CFX 17.0 computational fluid dynamics commercial software was used to solve the three-dimensional steady flow inside an axial-flow pump. The discretization of governing equations was based on the finite volume method, which considers the control volume composed of the internal cell nodes. The shear stress transport (SST) k-ω model was selected as the turbulence model to more accurately predict flow separation under an adverse pressure gradient. Compared with the standard and renormalization group k-ε models, this model considers the viscosity of the inner wall of the model. Therefore, it provides better turbulent shear stress transmission, a more stable algorithm, and better flow simulation performance in a narrow space [23,24]. The governing equations are as follows:

$$\frac{\partial(\rho k)}{\partial t} + \frac{\partial(\rho k u_i)}{\partial x_j} = \frac{\partial}{\partial x_j}\left[(\mu + \frac{\mu_t}{\sigma_k})\frac{\partial k}{\partial x_j}\right] + P_k - \beta' \rho k \omega \tag{3}$$

$$\frac{\partial(\rho \omega)}{\partial t} + \frac{\partial(\rho \omega u_i)}{\partial x_j} = \frac{\partial}{\partial x_j}\left[(\mu + \frac{\mu_t}{\sigma_{\omega 3}})\frac{\partial \omega}{\partial x_j}\right] - \beta \rho \omega^2 + \alpha_3 \frac{\omega}{k} P_k + 2(1 - F_1)\rho \frac{1}{\sigma_{\omega 2}\omega}\frac{\partial k}{\partial x_j}\frac{\partial \omega}{\partial x_j} \tag{4}$$

*2.2. Interphase Force Model*

The main interphase forces involved in the calculation of flow fields for a multiphase pump are resistance, lift, virtual mass, and turbulent dispersion. Resistance is produced by the relative motion of a gas and liquid phase and is the main cause of friction loss. Lift causes the gas phase to move perpendicular to the mainstream, which interferes with the mainstream and increases resistance. Virtual mass force is produced by local acceleration, and its effect on bubbles is opposite to that of centrifugal force. We assumed that the mainstream is perpendicular to the gravity, causing the bubbles move toward the rim before gathering at the hub in the impeller. Turbulent dissipative force is generated by the momentum exchange caused by the turbulence pulsation between phases. Its magnitude depends on the volume fraction gradient of the two phases [25,26]. The interphase force is shown in Figure 1 [27]. The net force of the interphase forces is $M_k$. The expression of the force in each phase is expressed as:

$$F_{M,k} = F_{D,k} + F_{A,k} + F_{L,k} + F_{T,k} \tag{5}$$

where $F_{D,k}$, $F_{A,k}$, $F_{L,k}$, and $F_{T,k}$ are the resistance, virtual mass, lift, and turbulent dispersion forces, respectively. The resistance force can be expressed as:

$$F_{D,g} = F_{D,l} = \frac{3}{4}\alpha_g \rho_l \frac{C_D}{d}|U_g - U_l|(U_g - U_l) \tag{6}$$

where $d$ is the bubble diameter; $U_g$ and $U_l$ are the absolute velocities in the gas and liquid phases, respectively; $\alpha_g$ is the volume fraction of the gas; $\rho_l$ is the liquid density; and $C_D$ is the resistance coefficient [28], which is expressed as:

$$C_D = \begin{cases} 24(1 + 0.15\text{Re}^{0.687})/\text{Re} & \text{Re} \leq 1000 \\ 0.44 & \text{Re} > 1000 \end{cases} \tag{7}$$

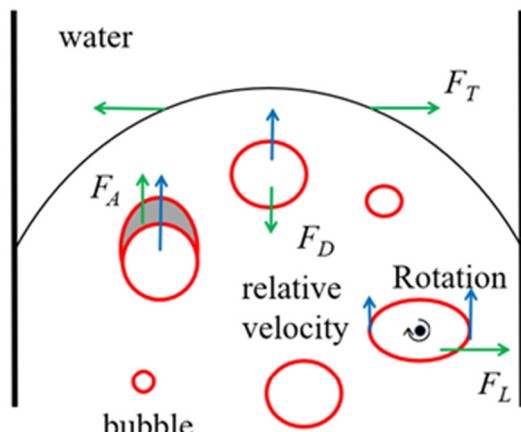

**Figure 1.** Interphase forces of gas–liquid flow in a multiphase pump [27].

The virtual mass force is expressed as:

$$F_{A,g} = F_{A,l} = \alpha_g C_A \rho_l \left( \frac{DU_g}{Dt} - \frac{DU_l}{Dt} \right) \tag{8}$$

where $C_A = 0.5$ is the virtual mass-force coefficient [26].

The lift force is expressed as:

$$F_{L,g} = F_{L,l} = C_L \alpha_g \rho_l (U_g - U_l) \times (\nabla \times U_l) \tag{9}$$

where $C_L = 0.5$ is the lift coefficient [27].

The turbulent dispersion force is expressed as:

$$F_{T,g} = F_{T,l} = -C_T \rho_l k \nabla \alpha_l \tag{10}$$

where $C_T = 0.1$ is the coefficient of turbulent dispersion force [20], and $k$ is turbulent kinetic energy.

*2.3. Computational Model*

The axial-flow multiphase pump model includes an inlet pipe, outlet pipe, impeller, and diffuser, as shown in Figure 2. The main design parameters are as follows. The design head is $H_d = 15$ m, design flow is $Q_d = 50$ m$^3$/h, and rated speed is $n = 2950$ r/min. The structural parameters are listed in Table 1.

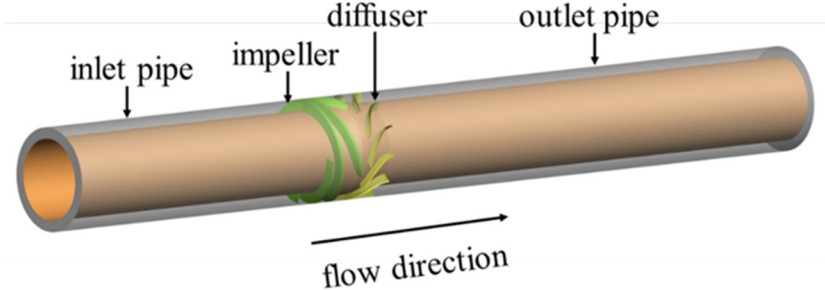

**Figure 2.** Computational model of an axial-flow multiphase pump.

**Table 1.** Structural parameters.

| Parameter | Impeller | Diffuser |
|---|---|---|
| Blade number | 4 | 11 |
| Shroud radius (mm) | 75 | 75 |
| Inlet hub radius (mm) | 58.96 | 67 |
| Outlet hub radius (mm) | 67 | 60 |
| Axial length (mm) | 55 | 65 |
| Tip clearance (mm) | 0.3 | 0 |

All flow components adopted a hexahedral structured grid, considering the full flow-channel structure and computer configuration (Intel Xeon W-2223 @ 3.6 ghz with 32 GB RAM). Integrated computer-aided engineering and manufacturing was adopted for the simple structure of the inlet and outlet pipes. The impeller and diffuser flow-channel structure was complex, so TurboGrid mesh generation software was used to divide the hexahedron structure grid. The target passage mesh size method was selected for global mesh partitioning. The wall area was encrypted with the proportional to mesh size method. The meshes of the impeller and diffuser are shown in Figure 3.

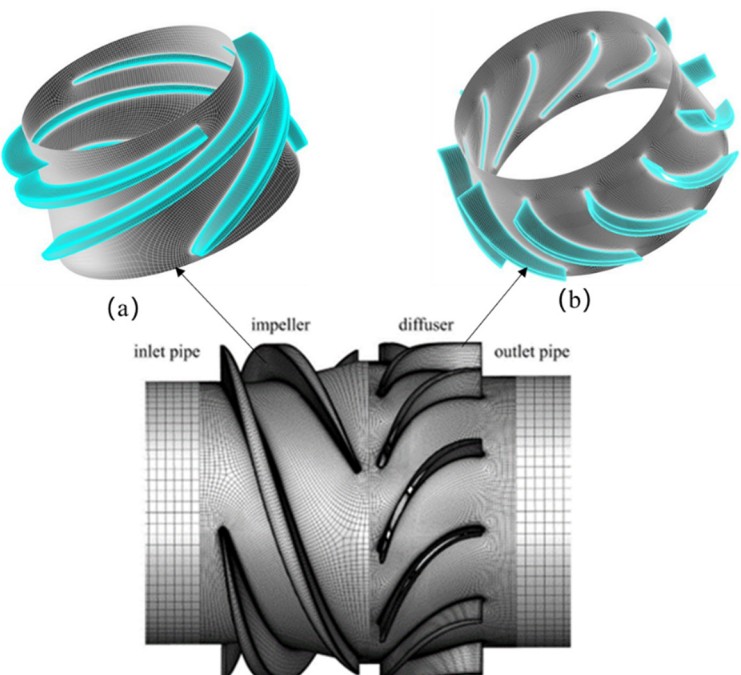

**Figure 3.** Computational meshes of the (**a**) impeller and (**b**) diffuser.

Grid independence verification analysis was conducted on the premise of ensuring grid quality. Five schemes with different grid numbers were verified and analyzed under a pure water condition (IGVF = 0%) with head and efficiency as the judgment criteria. Considering computing resources and accuracy, Scheme 3 had a grid number of 3,412,466 and was selected for numerical calculations. The results showed that the number of grids exceeded that of Scheme 3, and the head and efficiency changed gradually with the number of grids, as shown in Figure 4. The inlet section, impeller, diffuser, and outlet section had 143,658, 1,604,584, 1,478,546, and 185,678 grids, respectively.

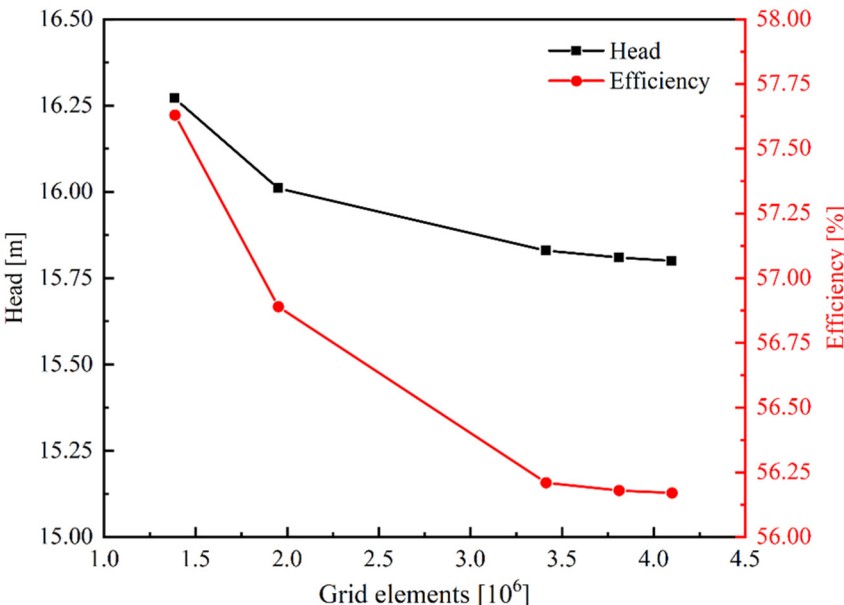

**Figure 4.** Verification of rid independence.

### 2.4. Boundary Conditions and Solution Settings

The high-resolution and second-order backward Euler models were selected because the SST k-ω turbulence model was used to solve the flow state inside the multiphase pump. An air–water two-phase mixed medium was selected as the flow medium. Water was the continuous phase, air was the discrete phase, and the bubble diameter was set as 0.1 mm [12]. The bubble diameter was assumed to be constant during the calculation. The volume fractions of air and water were determined according to working conditions. In all cases, they must add up to 1. The surface tension coefficient was set to 0.072 N/m, and interphase transfer was used as the particle model. The appropriate boundary conditions were set according to the operating conditions. The inlet was set to mass flow, and the outlet condition was the average static pressure outlet. The speed was set to 2950 r/min. The impeller was in the rotation domain, and the other regions were in the rest domain. The interfaces between the inlet pipe and impeller and between the impeller and diffuser adopted a frozen rotor interface. Other interfaces were set to general connection.

## 3. Numerical Results and Discussion

### 3.1. Analysis of External Characteristics

The influence of IGVF on the pressure rise performance of a multiphase pump was analyzed, as shown in Figure 5. The pressure rise was approximately linear with IGVF. When IGVF increased, the pressure rise decreased; that is, the pressurization capacity of the multiphase pump gradually decreased.

The gas-phase distributions in the flow passage of the impeller and diffuser are shown in Figure 6. The gas in the impeller runner gathered at the hub, mainly due to the density difference between gas and liquid. The gas was mainly concentrated in the hub area due to the high-speed rotation of the impeller. At the inlet of the diffuser, gas mainly collected at the hub. The gas gradually diffused toward the rim along the flow direction. The main reason for this is that the fluid maintained a velocity component in the circumferential direction due to inertia at the inlet. This left the fluid subject to centrifugal force, causing gas to accumulate at the hub. The inertial effect diminished as the fluid gradually moved toward the outlet. Subsequently, the fluid was less affected by centrifugal force, the degree of gas gathering at the hub gradually weakened, and the gas slowly diffused from the hub to the rim. In addition, the degree of gas accumulation in the impeller and diffuser passage gradually increased with an increase in IGVF. This is the main reason for the weakening of the supercharging capacity of the multiphase pump. The power capacity of the impeller

and diffuser gradually decreased with the increasing degree of gas accumulation in the impeller and diffuser.

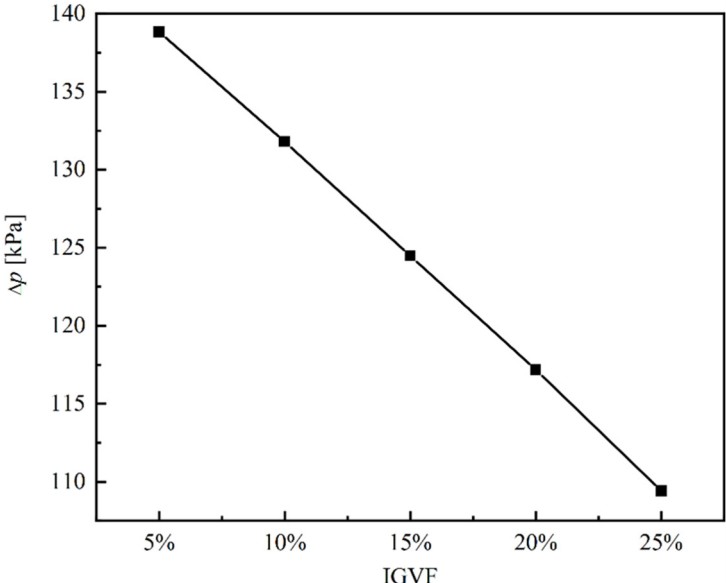

**Figure 5.** Influence of IGVF on pressure rise of a multiphase pump.

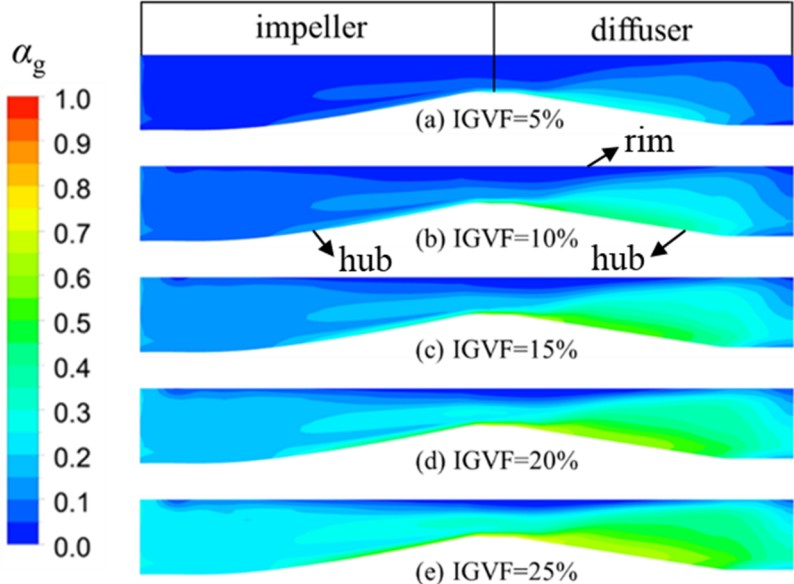

**Figure 6.** Distributions of gas void fractions in the impeller and diffuser under different IGVFs.

Next, the influence of flow rate on the pressure rise performance of the multiphase pump was analyzed. The pressure rise first increased and then decreased with an increase in flow rate, as shown in Figure 7. The reason for this is that the flow angle at the inlet of the diffuser was inconsistent with that of the blade, resulting in deviation from the design condition, as shown in Figure 8.

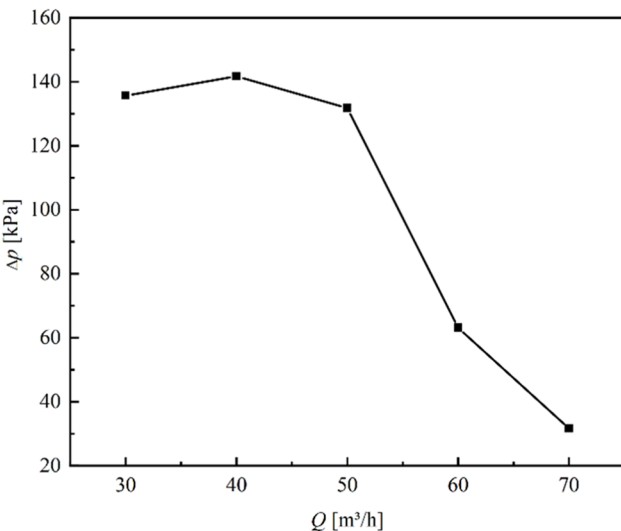

**Figure 7.** Influence of flow rate on pressure rise of a multiphase pump.

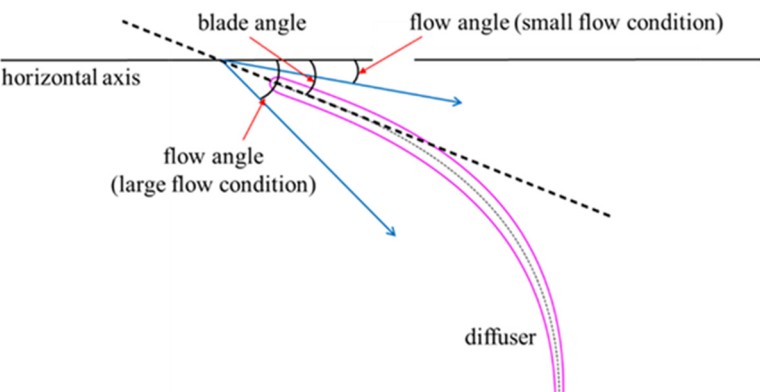

**Figure 8.** Diagram of flow angle.

The internal flow of the multiphase pump under different flow conditions was analyzed using a blade height of 0.5 as an example, as shown in Figure 9. The flow angle was smaller than the blade placement angle under the small low-flow condition. This tended to form shedding vortices on the blade suction surface, and the position of the shedding vortex was delayed as the flow increased. A shedding vortex easily formed on the blade pressure surface under a high flow condition because the streamline distribution was chaotic, as shown in Figure 9. Therefore, the flow angle at the diffuser inlet was inconsistent with the blade when deviating from the design condition. Subsequently, it was easy to form a shedding vortex and cause energy loss at the inlet of diffuser. Compared with low flow, the streamline distribution was more chaotic, and the energy loss was more serious under a high flow condition. Therefore, the pressurization capacity clearly decreased under the condition of high flow.

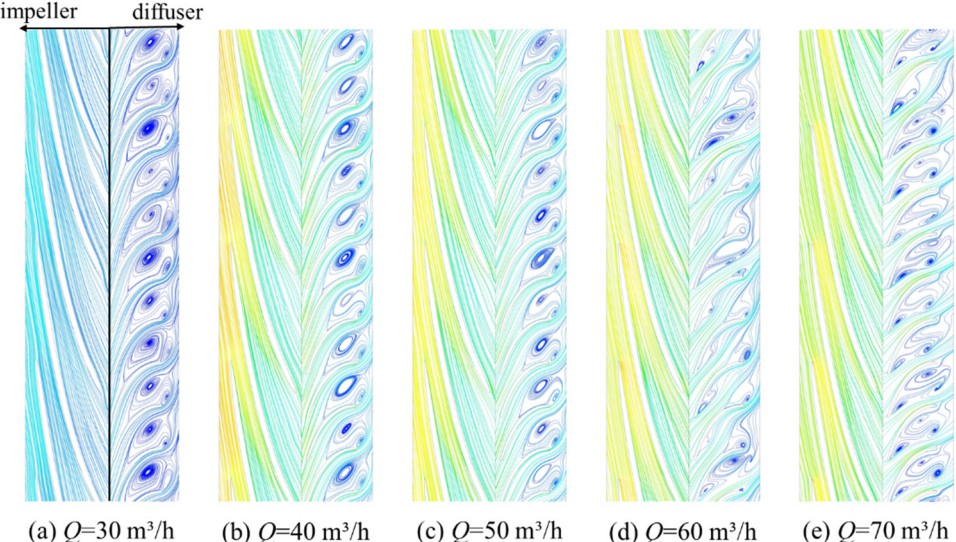

(a) *Q*=30 m³/h  (b) *Q*=40 m³/h  (c) *Q*=50 m³/h  (d) *Q*=60 m³/h  (e) *Q*=70 m³/h

**Figure 9.** Streamline distribution in the impeller and diffuser at span = 0.5 for different flow rates (*Q* = m³/h).

### 3.2. Influence of IGVF on Interphase Force

The change rule of interphase force under different IGVF conditions (*Q* = 50 m³/h) is shown in Figure 10. IGVF exerted a considerable influence on resistance and lift but had almost no influence on virtual mass and turbulent dispersion forces. The largest interphase force was drag force, followed by lift, virtual mass, and turbulent dispersion forces. Compared with other forces, the turbulent dispersion force was negligible. There were three peaks of interphase force along the flow direction, with a maximum value from the outlet of the impeller to the inlet of the diffuser; that is, the interaction force was the maximum in the area of static and static interference. The interphase force also reached its peak value in the impeller inlet and diffuser outlet areas. The main reason for this is that the interaction force was larger in the area of rotor–stator interference. In other regions, the IGVF was smaller, gas accumulation in the flow passage was not obvious, and the flow pattern was good. Therefore, the variation trend of interphase forces was also relatively orderly. With an increase in IGVF, the order of interphase forces was drag, lift, virtual mass, and turbulent dispersion. The variation trend of each force along the axial relative position remained basically unchanged, but the overall trend was upward. In particular, the range of change was large near the impeller inlet and diffuser outlet. The main reason for this is that the fluid in the impeller inlet entered the impeller runner from the inlet pipe. The high-speed rotation of the impeller led to fluid impingement on the leading edge of the blade, resulting in a chaotic flow field at the impeller inlet. A large amount of gas accumulated near the diffuser outlet to form an air sac, as shown in Figure 6. Resistance, virtual mass, and lift forces were positively correlated with gas holdup, according to the expression of the interphase force. Therefore, the interaction force was larger, and the change rule was more complex.

Contour plots of turbulent kinetic energy distribution in the impeller and diffuser passage under different IGVF conditions are shown in Figure 11. The turbulent kinetic energy in the guide-blade passage was significantly greater than that in the impeller passage, indicating that the flow pattern in the guide-blade passage was more chaotic. The turbulent kinetic energy in the impeller passage near the hub area increased with an increase in IGVF, whereas the change in turbulent kinetic energy in the diffuser passage was more obvious. This result is consistent with the streamline distribution in the flow channel shown in Figure 9. This indicates that the increase in IGVF led to an increase in turbulent kinetic energy and energy loss in the impeller and guide-blade runner. The interphase force also increased with an increase in IGVF according to the above analysis of

interphase force. The increase in interphase force led to an increase in energy loss in the impeller and diffuser passage.

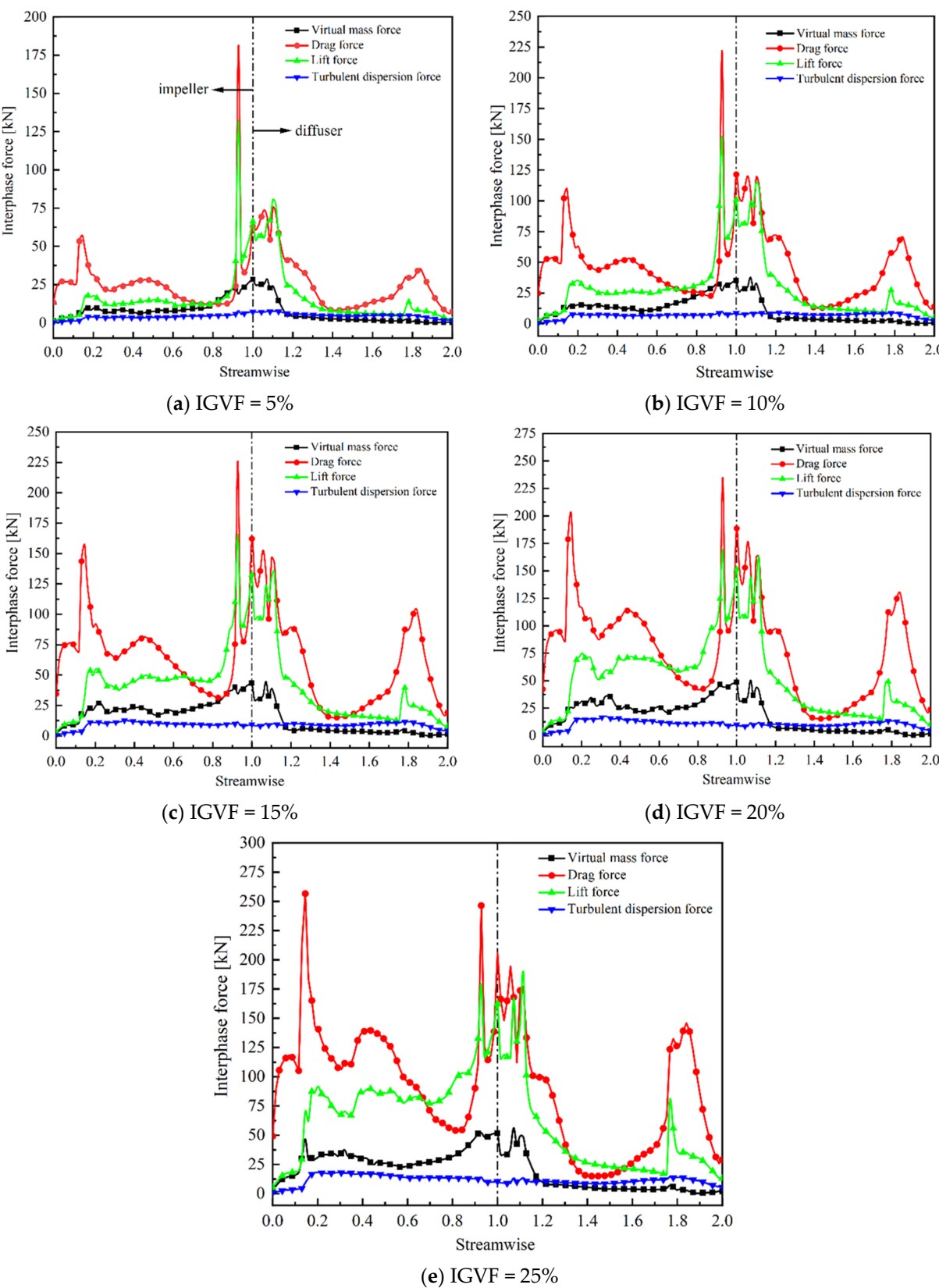

**Figure 10.** Analysis of interphase force under different IGVF conditions.

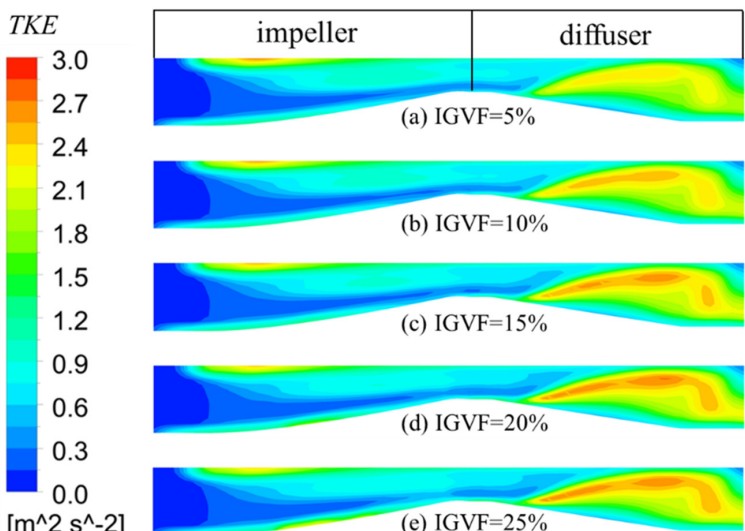

**Figure 11.** Distribution of turbulence kinetic energy in the impeller and diffuser under different IGVFs.

*3.3. Influence of Flow on Interphase Force*

The change rule of interphase force under different flow conditions with IGVF = 10% is shown in Figure 12. A gas holdup distribution of the impeller and diffuser passage under different flow conditions with IGVF = 10% is shown in Figure 13. Flow had a considerable influence on resistance and lift but almost no influence on virtual mass and turbulent dispersion forces, as shown in Figure 12. The order of the interphase forces was drag, lift, virtual mass, and turbulent dispersion. The resistance and lift reached the maximum near the impeller inlet when $Q = 30$ m$^3$/h, followed by the region of rotor–stator interference. It was evident that resistance and lift were affected by the gas holdup in the flow passage. The gas content in the impeller passage was high when $Q = 30$ m$^3$/h, resulting in significantly increased resistance and lift relative to other areas, as shown in Figure 13. The maximum resistance and lift forces were between the impeller outlet and guide-blade inlet when $Q \geq 40$ m$^3$/h. This was affected by static and static interference, resulting in increased drag and lift. Additionally, the impeller inlet and diffuser outlet resistance and lift were relatively large. The interphase force was larger, mainly due to the collision between the high-speed rotation of the impeller and the fluid at the inlet of the impeller. The outlet position of the diffuser was affected by the inconsistency between the inlet flow angle of the diffuser and blade placement angle. This created a vortex at the outlet of the diffuser that caused the interaction force to be high. The gas content in the impeller and diffuser passage dropped significantly and tended to be evenly distributed when $Q \geq 40$ m$^3$/h, with a minimal impact on the interphase force.

The designed flow rate in this study was Q = 50 m$^3$/h, and IGVF was set to 10% when calculating each flow condition. The corresponding interphase force under the condition of IGVF = 10% and $Q = 50$ m$^3$/h was used as the benchmark to more clearly demonstrate the influence of flow rate on interphase force. The relative size of the interphase force under other working conditions was compared, namely for F/F (IGVF = 10%, $Q = 50$ m$^3$/h). Here, the numerator F is the interphase force under different working conditions, and the denominator F (IGVF = 10%, $Q = 50$ m$^3$/h) is IGVF = 10%, corresponding to an interphase force at $Q = 50$ m$^3$/h. The relative forces of each phase under different flow conditions are shown in Figure 14. The virtual mass force was less affected by the flow rate in the impeller passage, as shown in Figure 14a. However, the virtual mass force was considerably affected by the flow in the diffuser passage, especially when it deviated from the design condition ($Q = 50$ m$^3$/h). Additionally, it changed considerably under the condition of high flow rate ($Q = 70$ m$^3$/h). The resistance in the impeller passage was considerably affected by the condition of low flow ($Q = 30$ m$^3$/h), as shown in Figure 14b. The change was

gradual under the other flow conditions, but the resistance clearly fluctuated in the diffuser passage when the flow exceeded that of the design condition. The variation law of lift was similar to that of resistance, as shown in Figure 14c. The variation of turbulent dispersion force was complex, as shown in Figure 14d. The turbulent dispersion force achieved the maximum value and gradually decreased along the flow direction at the middle and rear of the impeller and inlet of the diffuser under the condition of low flow rate ($Q = 30$ m$^3$/h). The turbulent dispersion force in the diffuser was considerably affected by the flow with a high flow rate.

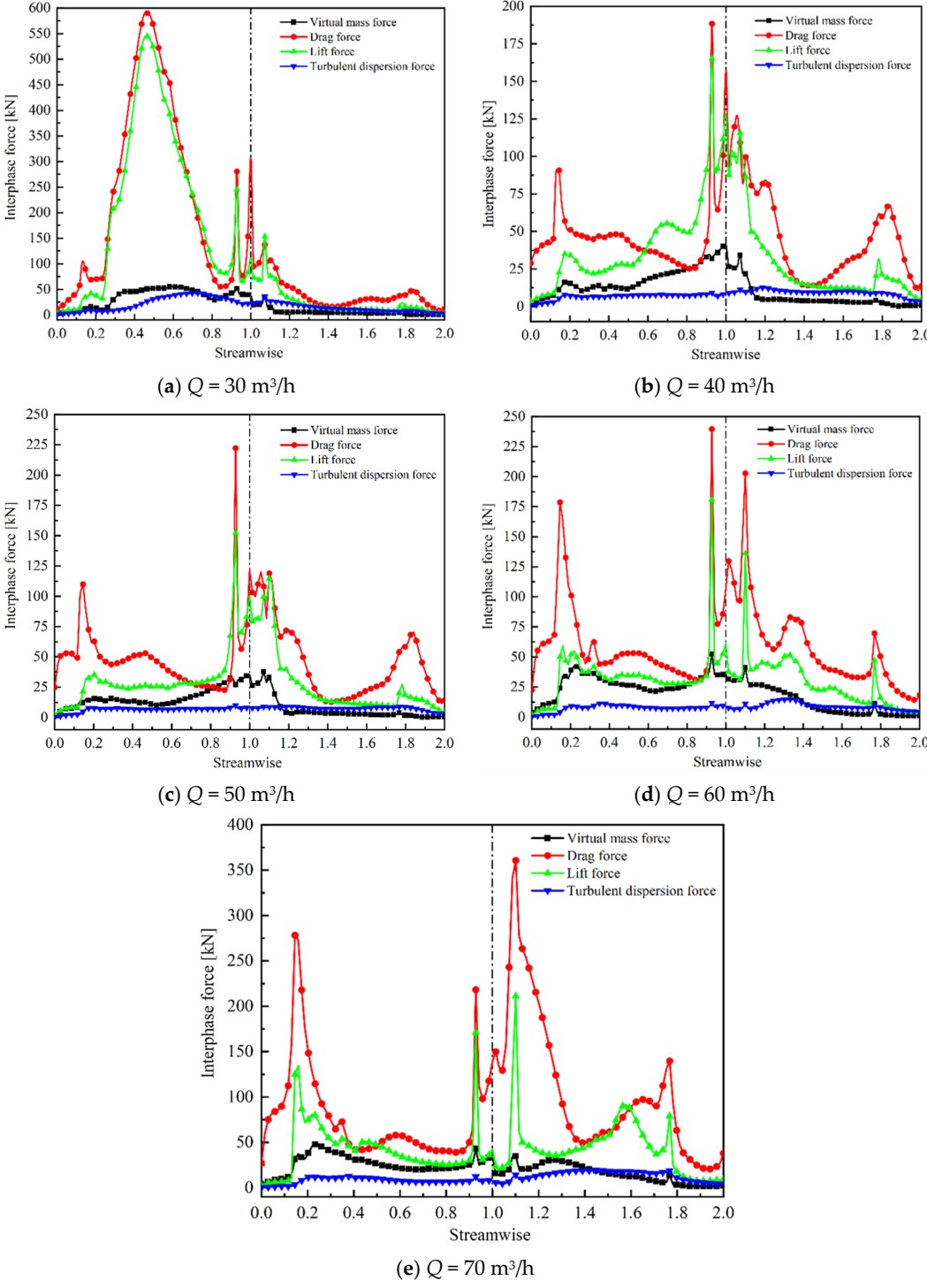

**Figure 12.** Analysis of interphase force under different flow rates ($Q = $ m$^3$/h).

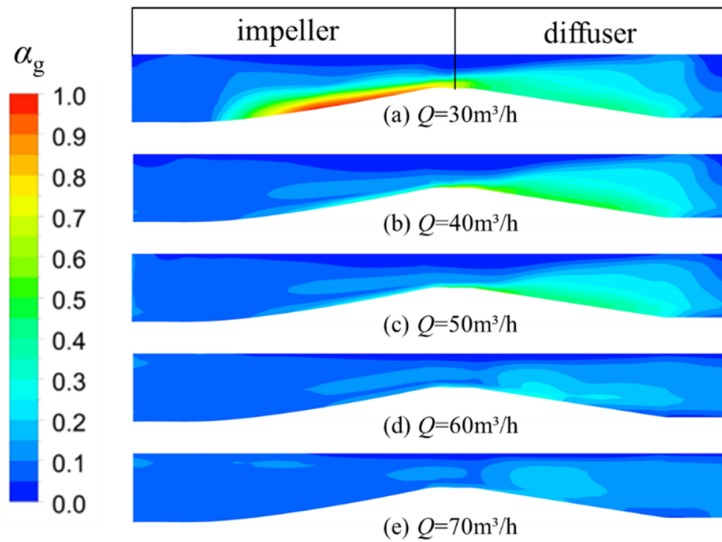

**Figure 13.** Distribution of gas void fraction in the impeller and diffuser under different flow rates ($Q$ = m$^3$/h).

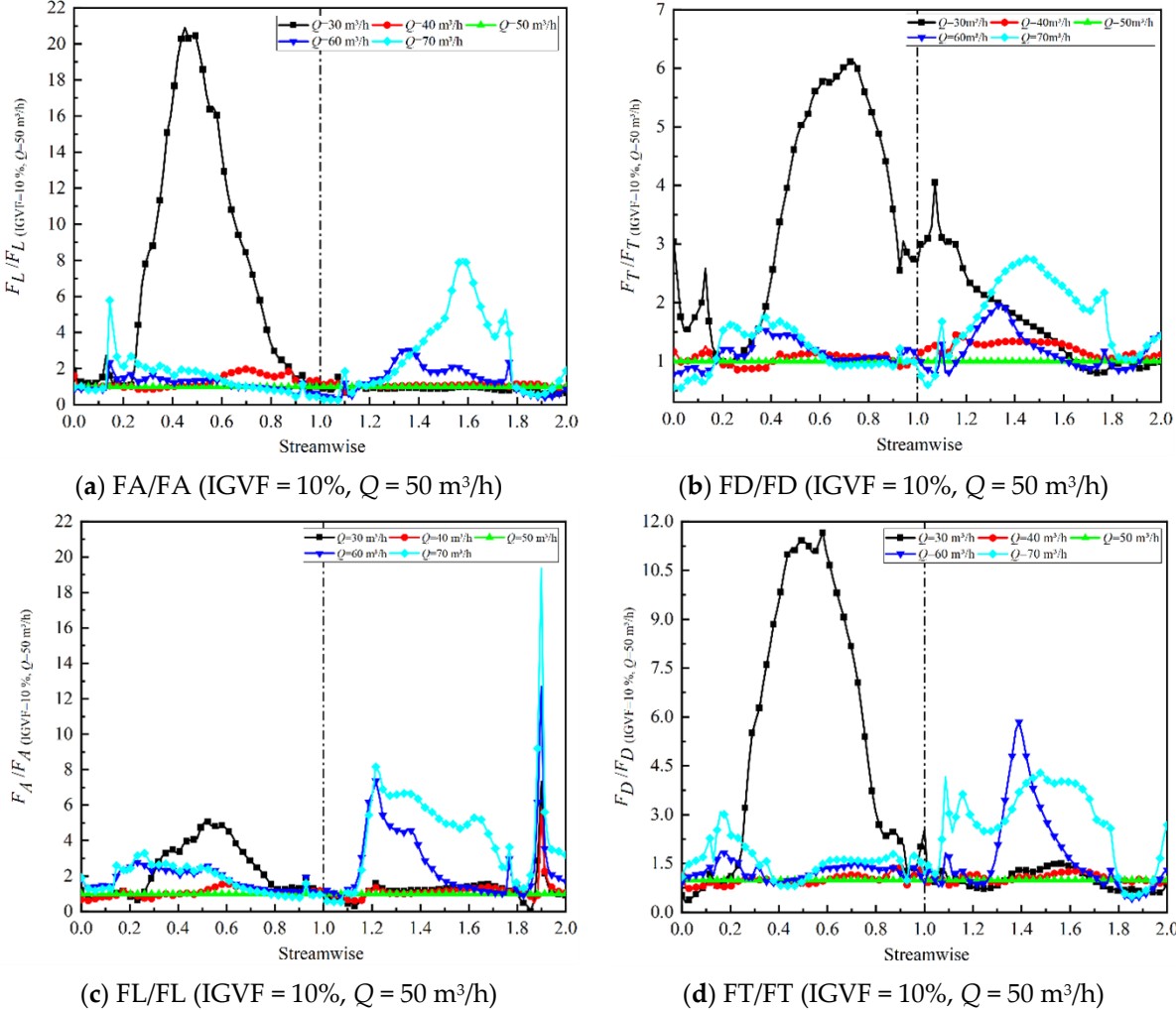

**Figure 14.** Magnitude ratio of interphase forces under different flow rates (IGVF = 10%, $Q$ = 50 m$^3$/h): (**a**) FA/FA, (**b**) FD/FD, (**c**) FL/FL, and (**d**) FT/FT.

The distribution of turbulent kinetic energy in the impeller and diffuser passage under different flow conditions is shown in Figure 15. The turbulent kinetic energy generates a local high-intensity region at the hub in the middle of the impeller runner when $Q$ = 30 m$^3$/h, as shown in area A in the black dotted box in Figure 15. This region corresponds to the position where the resistance, lift, and turbulent dispersion forces in the middle rotation path increased, as shown in Figure 14. When $Q$ > 30 m$^3$/h, the turbulent kinetic energy in the impeller and diffuser passage decreased with an increase in flow rate, and the boundary was $Q$ = 50 m$^3$/h. The change in turbulent kinetic energy in the impeller passage was no longer obvious when $Q$ > 50 m$^3$/h, but the turbulent kinetic energy in the diffuser passage gradually increased. The turbulent kinetic energy occurred in the middle and rear segments of the guide blade when $Q$ = 70 m$^3$/h, as shown in B in the black dotted box in Figure 15. This corresponds to the position at which the virtual mass force increased, as shown in Figure 14. The region of turbulent kinetic energy enhancement corresponded to the position of interphase force increase. This demonstrates that the increase in interphase force led to an increase in energy loss in the impeller and diffuser passage.

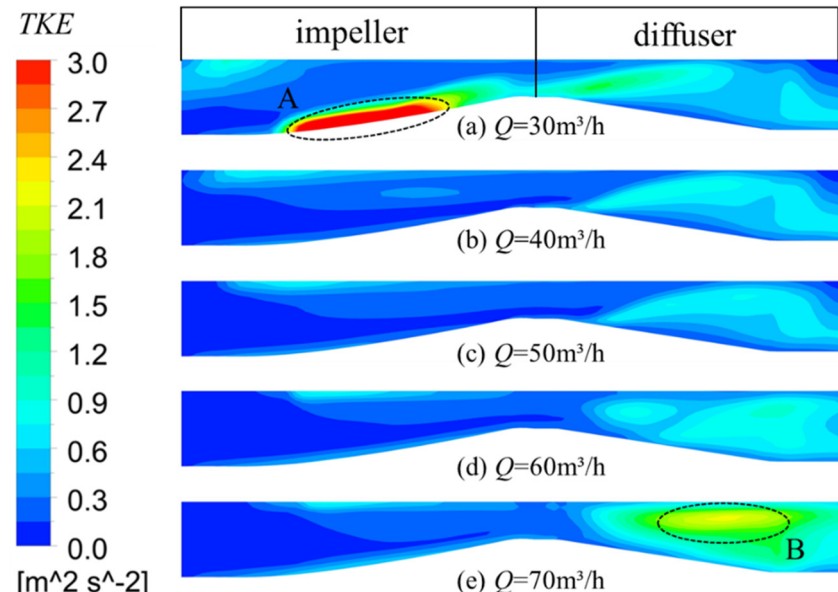

**Figure 15.** Distribution of turbulent kinetic energy in the impeller and diffuser under different flow conditions.

## 4. Conclusions

In this study, we analyzed gas–liquid interaction characteristics using a heterogeneous two-fluid model to investigate the influence of interphase force on multiphase pump performance. Full-channel numerical calculations were conducted, and the change rule of interphase force in an axial-flow multiphase pump was studied under different IGVF (5%, 10%, 15%, 20%, and 25%) and flow rate ($Q$ = 30, 40, 50, 60, and 70 m$^3$/h) conditions to illustrate their effects on interphase force. The following conclusions can be drawn:

1. Under different IGVF conditions, the pressure rise decreased with an increase in IGVF. Under different $Q$ conditions, the pressure rise first increased and then decreased with an increase in $Q$.

2. Under different IGVF conditions, the order of the interphase forces was drag, lift, virtual mass, and turbulent dispersion. The interaction force of each phase was relatively large in the region of static and static interference. The drag, lift, virtual mass, and turbulent dispersion forces all showed an increasing trend with an increase in IGVF. The interphase force increased considerably in the impeller passage but less in the guide-blade passage.

3. The largest interphase action order under different Q-flow conditions was drag, lift, virtual mass, and turbulent dispersion forces, which all tended to increase when deviating from the design condition. The force of each phase changed considerably in the impeller passage under the condition of low flow. Conversely, the force of each phase changed considerably in the diffuser passage under the condition of high flow rate.

4. The change trends of the interphase force and turbulent kinetic energy in the impeller and diffuser passage were basically constant with the changes in IGVF and flow conditions. This shows that the interphase force led to a change in turbulent kinetic energy, which affected the internal energy loss of the multiphase pump.

The results of this study provide important information for the optimization of the hydraulic design of multiphase pumps.

**Author Contributions:** Conceptualization, Y.D. and X.W.; methodology, Y.D.; writing—original draft preparation, J.X.; writing—review and editing, Y.L.; visualization, Y.Z.; supervision, Y.D.; software, C.K. All authors have read and agreed to the published version of the manuscript.

**Funding:** This research was funded by the Open Research Subject of the Key Laboratory of Fluid and Power Machinery, Ministry of Education (Grant No. LTDL2020-005), Doctoral Research Foundation of Lanzhou City University (Grant No. LZCU-BS2019-07).

**Institutional Review Board Statement:** Not applicable.

**Informed Consent Statement:** Not applicable.

**Data Availability Statement:** All relevant data are presented in the article.

**Acknowledgments:** The authors would like to thank all staff of the treatment plants for their contributions.

**Conflicts of Interest:** The authors declare no conflict of interest.

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
