# Peer review of "Gas–Liquid Interaction Characteristics in a Multiphase Pump under Different Working Conditions"

_processes, doi:10.3390/pr10101977_

Round 1
Reviewer 1 Report
Please see the attached file.

Author Response
Dear reviewer,
Thank you for your comments on our article. We have responded to your comments and revised the article. Our reply is attached below for your convenience.
Sincerely,
Yuxuan Deng
BaiLie School of Petroleum Engineering, Lanzhou City University Telephone:+8615908150884
dengyuxuan@lzcu.edu.cn

Reviewer 2 Report
The paper is very interesting, presents clearly and orderly. The obtained results are discussed in detail. Authors tried found the cause occurring phenomena. The form the article is correct.
1. Have the Authors compared results of numerical calculations with results of experimental studies?
2. Did the results differ much?
3. How does the gas-liquid interaction mechanism affect the performance of a multiphase pump?
The research is relevant and interesting.
The Authors were analyzed new factors, which influences on the work of pumps. Pomps are very important and popular devices used in chemical engineering. These results provide an important guiding significance to optimize the hydraulic design of multiphase pumps. Additionally, using the numerical calculations give new possibilities and reduces research time.
The paper is well written and text is clear and easy to read.
Conclusion is consistent with the evidence and arguments presented and they address the main question posed
Author Response

(The authors gave the same response as above.)

Round 2
Reviewer 1 Report
The authors properly addressed my questions and comments. Thanks.